# Degradation of Benzotriazole UV Stabilizers in PAA/d-Electron Metal Ions Systems—Removal Kinetics, Products and Mechanism Evaluation

**DOI:** 10.3390/molecules27103349

**Published:** 2022-05-23

**Authors:** Dariusz Kiejza, Joanna Karpińska, Urszula Kotowska

**Affiliations:** 1Doctoral School of Exact and Natural Sciences, University of Bialystok, Ciołkowskiego 1K St., 15-245 Białystok, Poland; d.kiejza@uwb.edu.pl; 2Department of Analytical and Inorganic Chemistry, Faculty of Chemistry, University of Bialystok, Ciołkowskiego 1K St., 15-245 Białystok, Poland; joasia@uwb.edu.pl

**Keywords:** benzotriazole UV stabilizers, peracetic acid, advanced oxidation processes, iron ions, cobalt ions

## Abstract

Benzotriazole UV stabilizers (BUVs) have gained popularity, due to their absorption properties in the near UV range (200–400 nm). They are used in the technology for manufacturing plastics, protective coatings, and cosmetics, to protect against the destructive influence of UV radiation. These compounds are highly resistant to biological and chemical degradation. As a result of insufficient treatment by sewage treatment plants, they accumulate in the environment and in the tissues of living organisms. BUVs have adverse effects on living organisms. This work presents the use of peracetic acid in combination with d-electron metal ions (Fe^2+^, Co^2+^), for the chemical oxidation of five UV filters from the benzotriazole group: 2-(2-hydroxy-5-methylphenyl)benzotriazole (UV-P), 2-tert-butyl-6-(5-chloro-2H-benzotriazol-2-yl)-4-methylphenol (UV-326), 2,4-di-tert-butyl-6-(5-chloro-2H-benzotriazol-2-yl)phenol (UV-327), 2-(2H-benzotriazol-2-yl)-4,6-di-tert-pentylphenol (UV-328), and 2-(2H-benzotriazol-2-yl)-4-(1,1,3,3-tetramethylbutyl)phenol (UV-329). The oxidation procedure has been optimized based on the design of experiments (DoE) methodology. The oxidation of benzotriazoles follows first order kinetics. The oxidation products of each benzotriazole were investigated, and the oxidation mechanisms of the tested compounds were proposed.

## 1. Introduction

UV stabilizers are, widely, used to prevent the degradation of polymeric products. UV radiation has a destructive effect on the structure of plastics, by changing the mass, color, gloss, and other properties, leading to a constant demand for compounds protecting them against UV degradation. As a result of their massive production, the global UV absorbers market is expected to reach a value of USD 669 million in 2020 and USD ~920 million by 2027 [1]. Benzotriazoles (BTRs) constitute a vast group of heterocyclic chemicals, including those with anti-corrosive and anti-icing properties as well as UV stabilizers (BUVs). The last group of BTRs are compounds containing an additional phenolic ring attached to the benzotriazole molecule (Figure 1).

BUVs are used in the following products and fields: adhesives, engineering thermoplastics (UV-P), engineering plastics, sealants, wood lacquers (UV-326), coatings, fibers, films, lacquers (UV-327), automotive and industrial coatings, carpets, enamels, packaging, paints, photographic coating, plastics, stain, textiles, thermosetting acrylic enamels (UV-328), gel coats, glazing materials, marine and auto applications, molded articles, photoproducts, sheets, and signs (UV-329) [2]. The practical use of benzotriazoles dates back to the early 1960s, when the first patent for the use of benzotriazoles, as stabilizers for polymer products, was obtained [3]. BUVs are capable of absorbing UV light (300–400 nm) [4], and their protective properties have, also, used the production of cosmetics [5,6]. Benzotriazoles are expected to dominate the market for UV filters, due to their excellent spectral coverage and high molar extinction coefficient [1]. According to the OECD Existing Chemicals Database, UV-P, UV-328, and UV-329 are designated as high production volume chemicals (HPVC), with production >1000 tons per year [7]. Based on the SPIN database (Substances In Preparations In Nordic Countries), the total use of BUVs in Denmark, Finland, Norway, and Sweden are 0.4–11.6, 4–10, 0.1–0.9, 0.1–4.5, and 0.6 t/a for UV-P, UV-326, UV-327, UV-328, and UV-329, respectively [8].

Benzotriazoles are a serious environmental threat, due to their high resistance to both biological and chemical degradation. BUVs end up in the environment, as a result of direct rain runoff and snowmelt [9]; however, the largest contamination source is run-off from sewage treatment plants [10,11,12,13,14,15,16]. Due to their resistance to the degradation processes, benzotriazole-based UV absorbers accumulate and persist in the environment. The presence of benzotriazoles in aquatic environment, such as marine organisms, river and lake waters, and sediments, is a major threat [17,18,19,20,21,22,23]. In addition, UV-P, UV-326, UV-327, and UV-328 were detected in house dust [24,25]. BUVs were also determined in PM10 outdoor air, from industrial areas in Tarragona, Spain [26]. Benzotriazoles trends to accumulate in the tissues of living organisms [22,27,28,29,30,31] and were detected in human breast milk [32,33,34]. Exposure to benzotriazoles via food packaging seems insignificant but is, potentially, harmful. Dietary exposure to benzotriazoles, based on maximum concentrations in foods (ng/kg bw/day), are 9.5–29.7 and 38.7–120.4 for UV-326 and UV-329, respectively [35]. The concentrations of BUVs in various samples are presented in Table 1.

Benzotriazoles raise concerns about the safety of living organisms and their surrounding environment. Only two of the benzotriazoles studied in this work are regulated by law. The European Chemicals Agency (ECHA) has classified UV-327 and UV-328 as substances with high bioaccumulation potential and strong resistance to degradation. UV-327 is on the list of substances considered as persistent organic pollutants (POPs). The Japanese government, also, classified UV-327 as a Monitoring Chemical Substance because of its high bioaccumulative characteristics [37]. The 16th meeting of the Persistent Organic Pollutants Review Committee of the Stockholm Convention (January 2021) concluded that UV-328 satisfies all the criteria to be included in the convention’s Annex D (Screening criteria for persistent organic pollutants), namely due to is persistence, bioaccumulation, potential for long-range environmental transport, and adverse effects on humans and/or the environment [38].

All benzotriazoles presented in this study raise doubts about their safety because they may contribute to adverse antiandrogenic effects [39]. Feng et al. [40] has proven that UV-P shows partial estrogenic activity against the human breast cancer MVLN cell line, while UV-329 is not estrogenic. In vitro experiments with human liver microsomes (HLMs) were performed, to identify the phase I metabolites of UV-327 and UV-328, which can be used as potential biomarkers for exposure to these compounds [41,42]. UV-328 metabolites are, also, detected in human urine and blood [43,44]. Detection of metabolites can elucidate the pathway of metabolism and estimate the toxicity of specific metabolism products towards living organisms. The UV-328 and UV-P metabolites have greater antiandrogenic activity upon human CYP3A4-mediated biotransformation than their non-metabolized forms [45]. UV-328, adversely, affects the thyroid hormone pathway of the zebrafish, Danio rerio [46]. Chronic exposure to low concentrations of this UV stabilizer causes oxidative stress and liver damage in zebrafish [47]. Freshwater green algae *Chlamydomonas reinhardtii* subjected to UV-328 show increased production of reactive oxygen species, while prolonged contact with UV-234 caused an increase in lipid peroxidation [48]. Knowledge about concentrations that cause negative effects on living organisms is limited, with only a few studies available for the toxicity assessment of BUVs (Table 2.).

The aim of the presented work was to develop a new approach to the removal of BUVs as persistent environmental pollutants from aqueous solutions, based on the use of advanced oxidation. The literature review shows that, until now, no attempt has been made to use chemical oxidants to degrade and dispose of BUVs. This work presents the study on the oxidation process of five benzotriazole UV stabilizers that use peracetic acid activated with d-electron metal ions. Peracetic acid (PAA) is a long-known disinfectant and effective oxidant of organic micropollutants [53,54,55,56]. In recent studies, UV irradiation [57,58,59,60], d-electron metal ions [61,62,63,64], or heterogeneous activators [65,66,67,68,69,70,71] have been used as PAA activators. So far, no information is available on the removal procedures of benzotriazole UV stabilizers. Liu et al. [72] reported that sorption on sludge plays a dominant role in the removal of benzotriazole UV absorbers in municipal wastewater treatment plants. Chen et al. [73] described the photodegradation process of UV-P in coastal seawaters. The neutral form of UV-P is photodegraded more slowly than both the cationic and the anionic form. Singlet oxygen, hydroxyl radical, and dissolved organic matter have a positive effect on indirect UV-P photodegradation, in coastal seawaters. One of the latest reports appeared on the reductive photodegradation of BUVs in visible light, using tetraacetylated riboflavin (RFTA) as a photocatalyst [74].

## 2. Results and Discussion

### 2.1. Optimization of the UV Stabilizers Oxidation Process

Twenty experiments, including six repetitions under the same conditions in the central point, were carried out for three selected factors that affect oxidation efficiency at five levels. Table 3, Appendix A present data for the individual experiments as well as for the experimental and predicted values of removal efficiency (RE%). 

For each of the studied UV stabilizers (UV-P, UV-326, UV-327, UV-328, UV-329), a mathematical model was developed to characterize the relationship between the degradation efficiency, PAA, metal ion concentration, and pH of the solution. Statistical analysis of the developed regression model was performed for all benzotriazoles. ANOVA test results for fit UV-326 removal efficiency are presented in Table 4. Data for other benzotriazoles are included in the Appendix A.

The results obtained for UV-326 indicate that the regression model is characterized by a low coefficient of determination (R^2^ = 0.638). These results prove that the model only determines the influence of factors on the effectiveness of UV-326 removal in 64%.

Pareto charts (Figure 2, Appendix A) show a statistically significant (*p* < 0.05) influence of the individual independent variables on the UV stabilizers removal process. The presented diagrams show that the efficiency of micropollutants removal, most strongly, depends on the concentration of metal ions. In addition, the relationship between the concentration of the metal ion and the pH is noticeable because it determines the speciation of the activator necessary for the oxidation process initiation. In the Co^2+^/PAA process, all factors have a statistically insignificant effect on the removal efficiency of UV-327 and UV-328. This gives the information that any amount of oxidant and activator is good for these substances’ removal.

Figure 3, Appendix A present response surface plots of the removal efficiency of UV stabilizers depending on the combination of two independent variables: PAA concentration vs. pH, PAA concentration vs. metal ion concentration, and metal ion concentration vs. pH, with a predetermined value of the third variable. As can be seen in the graphs, there is no agreement as to the influence of a specific factor on the oxidation of benzotriazoles; however, a strong dependence of RE% on the concentration of PAA and the activator is observed. The higher the concentration of the oxidant and/or activator is, the greater the percent of degradation. This phenomenon can be explained by the fact that the number of radicals generated increases with an increase in PAA concentration. After exceeding the optimal concentration values, the oxidation efficiency may drop. It is observed that in the Fe^2+^/PAA process, the excess of PAA may interact with hydroxyl radicals, leading to the formation of radicals with lower reactivity [75]:CH_3_C(O)O_2_H + ^•^OH → CH_3_C(O)O_2_^•^ + CH_3_C(O)OH(1)
CH_3_C(O)O_2_H + ^•^OH → CH_3_C(O)^•^ + H_2_O + O_2_(2)
CH_3_C(O)O_2_H + ^•^OH → HO_2_^•^ + CH_3_C(O)OH(3)

There may be a situation where the excess PAA can react with acetoxyl radicals, resulting in the formation of less-reactive acetylperoxyl radicals:CH_3_C(O)O_2_H + CH_3_C(O)O^•^ → CH_3_C(O)O_2_^•^ + CH_3_C(O)OH(4)

For UV-327 and UV-328 in the Co^2+^/PAA process, no factor significantly contributes to the degradation efficiency. Various combinations of the independent variables result in the same oxidation value, of about 70%. It led to the conclusion that with any amount of oxidant and/or activator, and at any pH, it is possible to carry out the oxidation process. In both processes, the pH contributes to the efficiency of the UV stabilizer removal process. The pH of the system affects the radical formation process as well as the chemical forms of the oxidant and activators. Kim et al. [75] reported that PAA (pKa 8.2) at a pH of 3–7 exists in the form of neutral molecules. With increasing pH, the concentration of the ionized form increases. PAA^–^ shows weaker oxidizing properties than PAA^0^, but can react more easily with ^•^OH radicals, thus affecting the oxidation process. At a higher pH, Fe^2+^ can be oxidized more easily. Moreover, Fe^3+^ speciation is strongly dependent on pH; hydroxocomplexes may form or they may precipitate. The precipitated forms of iron are unable to activate the peracid which, in turn, reduces the concentration of radicals in the system.

Mechanisms of peracetic acid-based advanced oxidation processes are not, yet, well understood. The radicals formed as a result of PAA activation are not highly reactive but can be selective; therefore, optimization of the oxidation process brings different results for individually tested organic micropollutants. Nevertheless, optimization allows one to predict the success of the removal process. On the basis of the obtained results, it was found that the optimal conditions for the oxidation of UV stabilizers were C_PAA_ = 25 mg/L and C_Fe2+_ = 6·10^−4^ mol/L for the Fe^2+^/PAA process, and C_PAA_ = 40 mg/L and C_Co2+_ = 8 × 10^−4^ mol/L for the Co^2+^/PAA process. Both experiments were performed at a pH of 4.5.

### 2.2. UV Stabilizers Degradation Kinetics

Degradation of UV-P, UV-326, UV-327, UV-328, and UV-329 was investigated, at a pH of 4.5, in the Fe^2+^/PAA and Co^2+^/PAA systems. Initial benzotriazoles concentrations were 500 μg/L. In the Fe^2+^/PAA system, [Fe^2+^]_0_ = 6 × 10^−4^ mol/L and [PAA]_0_ = 25 mg/L, while in Co^2+^/PAA, [Co^2+^]_0_ = 8 × 10^−4^ mol/L and [PAA]_0_ = 40 mg/L. In the Fe^2+^/PAA process, 69%, 90%, 91%, 91%, and 89% degradation in 180 min was observed for UV-P, UV-326, UV-327, UV-328, and UV-329, respectively. UV-P is the least oxidized among all tested compounds. Similar results were obtained in the Co^2+^/PAA process, for the same time frame (180 min), where 96%, 95%, 91%, 89%, and 80% of UV-P, UV-326, UV-327, UV-328, and UV-329, respectively, were oxidized. In the case of UV-P, an immediate decrease in concentration was noticed within 5 min. This time is too short for the oxidation of the mentioned compound, since UV stabilizers are relatively difficult to degrade. This result can, potentially, be explained in two ways. In the Co^2+^/PAA system, either selective oxidation of UV-P takes place with only subsequent compounds created, or there is no oxidation process, with only the UV-P complexation reaction by cobalt ions [2,76]. The kinetics of tested compounds decomposition are presented in Figure 4 and Figure 5.

Pseudo-first order kinetic model was applied to determin the rate of tested organic micropollutants. First-order kinetic constants in the Fe^2+^/PAA process were 0.0059 min^−1^, 0.0118 min^−1^, 0.0166 min^−1^, 0.0125 min^−1^, and 0.0121 min^−1^, for UV-P, UV-326, UV-327, UV-328, and UV-329, respectively. Similar values were obtained in the Co^2+^/PAA process, where *k* was 0.0150 min^−1^, 0.0107 min^−1^, 0.0116 min^−1^, and 0.0088 min^−1^ for UV-326, UV-327, UV-328, and UV-329, respectively (Table 5).

### 2.3. Mechanism of UV Stabilizers Degradation

Commercial peracetic acid is, typically, an equilibrated mixture of PAA, hydrogen peroxide, acetic acid, and water, according to the reaction:CH_3_CO_2_H + H_2_O_2_ ↔ CH_3_C(O)O_2_H + H_2_O(5)

For this reason, activation of peracetic leads to the generation of reactive species involved in the oxidation of organic micropollutants. Using homogenous systems, such as UV irradiation andd-electron metal ions, ^•^OH, CH_3_C(O)OO^•^, CH_3_C(O)O^•^, and other radicals can be formed [57,58,61,62,75]. In the Fe^2+^/PAA process, Fe^2+^ reacts with PAA and H_2_O_2_. The reactions that take place are as follows [75]:CH_3_C(O)OOH + Fe^2+^ → CH_3_C(O)O^•^ + Fe^3+^ + OH^−^(6)
CH_3_C(O)OOH + Fe^2+^ → CH_3_C(O)O- + Fe^3+^ + ^•^OH(7)
CH_3_C(O)OOH + Fe^2+^ → CH_3_C(O)OH + Fe^IV^O^2+^(8)
H_2_O_2_ + Fe^2+^ → ^•^OH + Fe^3+^ + OH^−^(9)
H_2_O_2_ + Fe^2+^ → H_2_O + Fe^IV^O^2+^(10)

The main reactions taking place within the Co^2+^/PAA system are the formation of acetylperoxy (CH_3_C(O)OO^•^) and acetoxyl (CH_3_C(O)O^•^) from both cobalt species [61,62]:CH_3_C(O)OOH + Co^2+^ → CH_3_C(O)O^•^ + Co^3+^ + OH^−^(11)
CH_3_C(O)OOH + Co^3+^ → CH_3_C(O)OO^•^ + Co^2+^ + H^+^(12)

The mechanisms of the Fe^2+^/PAA and Co^2+^/PAA processes have been investigated, by determining the participation of individual radicals in the oxidation process of UV-stabilizers. To evaluate the ^•^OH radicals’ activity, tert-butyl alcohol (TBA) was used. The participation of O_2_^•−^ radical and ^1^O_2_ was also checked, by adding 1,4-BQ and NaN_3_ to the solution for superoxide anion radical and singlet oxygen, respectively. The influence of individual reactive species, on the removal efficiency of tested UV stabilizers in the Fe^2+^/PAA and Co^2+^/PAA systems, is shown in Figure 6 and Figure 7.

There are two main sources of hydroxyl radicals in the Fe^2+^/PAA process. ^•^OH radicals are formed from the direct reaction of PAA with iron ions and from the Fenton reaction taking place in the system, due to the presence of H_2_O_2_. The much higher reaction rate constant for the formation of radicals from peracetic acid at pH of 3.0–7.1 (0.5–1.10) × 10^5^ M^−1^∙s^−1^, compared to that of Fe^2+^/H_2_O_2_ (k = 63–76 M^−1^∙s^−1^), proves that PAA decomposition is the predominant source of ^•^OH radicals in Fe (II)/PAA systems [54]. As seen in Figure 1, the oxidation process of benzotriazoles takes place with a small share of hydroxyl radicals. Sodium azide, used as ^1^O_2_ quencher, increased the removal efficiency for almost all benzotriazoles. This phenomenon can be explained by the fact that at pH of 4–13, the N_3_^−^ ion can react with hydroxyl radicals to form azide radicals N_3_^•^, which, in turn, could oxidize organic compounds by electron transfer [77]:^•^OH + N_3_ → OH^−^ + N_3_^•^(13)

In the Co^2+^/PAA process, this phenomenon is not observed because fewer ^•^OH radicals that can oxidize azide are produced. This result may, indirectly, prove the presence of hydroxyl radicals in the iron (II)-activated PAA system. Superoxide anion radicals can, also, be formed in the system, as a result of the reaction accompanying the Fenton process:H_2_O_2_ + ^•^OH → HO_2_^•^ + H_2_O(14)
HO_2_^•^ → O_2_^•−^ + H^+^(15)

O_2_^•−^ may react with CH_3_C(O)OO^•^ [62], which removes radicals affecting the degradation process of UV stabilizers:CH_3_C(O)OO^•^ + O_2_^•−^ → CH_3_C(O)OO^−^ + O_2_(16)

The removal of superoxide anion, with 1,4-BQ from the system, increased the removal efficiency of most benzotriazoles. The use of tert-butyl alcohol does not cause any significant changes in the efficiency of benzotriazole removal, which proves the low participation of hydroxyl radicals in the Co^2+^/PAA process. ^•^OH are not generated in the direct reaction of PAA with Co^2+^ ions, although these can be formed from R-O^•^ radicals [62]. Singlet oxygen is produced by the decomposition of PAA [62,78]:CH_3_C(O)OOH + CH_3_C(O)OO^−(^→ CH_3_CH(O)OH + CH_3_C(O)O^−(^+ ^1^O_2_(17)

The effectiveness of BTA removal under the influence of singlet oxygen was assessed, by adding NaN_3_ to the reaction mixture. Sodium azide did not contribute to the slowing down of the UV stabilizers degradation process. It follows that singlet oxygen is formed in too small of an amount, which is not capable of efficient oxidation of organic micropollutants. HO_2_^•^/O_2_^•−^ radicals can be formed in the Co^2+^/PAA process, both from other radicals and through a reaction with hydrogen peroxide [62,75]:CH_3_C(O)OO^•^ → HO_2_^•^ + CH_2_CO(18)
H_2_O_2_ + CH_3_C(O)OO^•^ → HO_2_^•^ + CH_3_C(O)OOH(19)
CH_3_C(O)O^•^ + H_2_O_2_ → HO_2_^•^ + CH_3_C(O)OH(20)
CH_3_C(O)OO^•^ + H_2_O_2_ → HO_2_^•^ + CH_3_C(O)OOH(21)

As can be seen in the figure, superoxide anion has the greatest influence on the oxidation process of UV-326 and UV-327.

### 2.4. Benzotriazole UV Stabilizers Degradation Products

ESI-MS analysis of the oxidation products of the tested benzotriazoles was performed, and mass spectra were acquired (Appendix A). The registered spectra of post-reaction mixtures are similar because of the similar structures analyzed BUVs, therefore, some identical degradation products have been detected. On the other hand, the ions were characteristic, only for products made from only one of the benzotriazoles. Based on the ESI-MS spectra and the available literature data [79,80,81], the structures of the oxidation products have been proposed (Figure 8, Appendix A). Most of the structural changes occur within the benzotriazole ring, which was discussed in previous works. When comparing the oxidation products in the Fe^2+^/PAA and Co^2+^/PAA processes, no significant differences were noticed. Some products contain -OH groups, resulting from the action of the hydroxyl radical. It confirms its dominant role in the oxidation process. Generally, products with lower molecular weights are produced. This shows that the radicals break the molecules into smaller fragments and, probably, lead to complete mineralization.

## 3. Materials and Methods

### 3.1. Materials and Characterizaton

Benzotriazole UV stabilizers: UV-P, UV-326, UV-327, UV-328, UV-329 were obtained from Sigma-Aldrich (Steinheim am Albuch, Germany). Characteristics of UV stabilizers considered within this work are included in Appendix A. All standard reagents were at analytical grade. These were used to prepare a stock solution that contained 1 mg/mL of each chemical in acetonitrile, that was then stored at −18 °C, for no longer than one month. Working solutions, prepared by diluting the stock standard solution in acetonitrile, were stored at −18 °C, for no longer than two weeks. Chromatography-grade pure acetonitrile, supplied by Merck (Darmstadt, Germany), was used. Chromatography-grade pure chlorobenzene, produced by Sigma-Aldrich (Steinheim am Albuch, Germany), was applied as the extraction solvent. Ferrous sulfate heptahydrate (FeSO_4_·7H_2_O) (Chempur, Piekary Śląskie, Poland) and cobalt sulfate (CoSO_4_) (Sigma Aldrich, Steinheim am Albuch, Germany) were used to activate peracetic acid. Peracetic acid was synthesized on site, according to the procedure described in Appendix A. For this purpose, pure p.a. acetic acid, hydrogen peroxide (Chempur, Piekary Śląskie, Poland), and sulphuric acid (POCH, Gliwice, Poland) were used. Sodium thiosulfate (Na_2_S_2_O_3_), obtained from Thermo Fisher Scientific (Dreieich, Germany), was used as a radical scavenger. Tert-butyl alcohol TBA and sodium azide NaN_3_, acquired from Fisher Scientific (Merelbeke, Belgium), and 1,4-benzoquinone 1,4-BQ (Acros Organics, Geel, Belgium) were used to study the reaction mechanisms. Deionized water from the Milli-Q RG (Millipore, Burlington, MA, USA) purification system was stored in glass bottles.

Determination of the UV stabilizers in water samples was performed by gas chromatography with mass spectrometry (GC-MS). The chromatographic analysis was carried out, using a 7890B gas chromatograph with an electronic pressure control, and was coupled with a mass selective detector 5977A (electron impact source and quadrupole analyzer) from Agilent Technologies, USA. This device was equipped with an HP-5MS column (5% phenylmethylsiloxane), with dimensions of 30 m × 0.25 mm × 0.25 µm film thickness. Helium (99.999%), at a constant flow rate of 1.0 mL/min, was used as a carrier gas. An injector worked in splitless mode, at a temperature of 250 °C. The oven was operating on the following temperature schedule: start at 120 °C, raise the temperature by increments of 10 °C/min, until reaching 290 °C, then, by increments of 20 °C, until reaching 310 °C. Each temperature was maintained for 1 min, for a total run time of 19 min. The electron impact source temperature was 230 °C, with electron energy of 70 eV. The quadrupole temperature was 150 °C, and the GC interface temperature was 280 °C. The MS detector was set to work in selected ion monitoring (SIM) mode. Target compound monitored ions are shown in Appendix A. Then, calibration curves were prepared to calculate BUVs’s concentrations during the oxidation reaction (Appendix A).

In order to identify the oxidation products, analyses were performed using an Agilent 6530 Accurate-Mass Q-TOF ESI (+) and LC/MS system, equipped with an Agilent Poroshell 120 EC-C18 column (2.7 μm × 3.0 × 150 mm). The gradient mobile phase was A: water, B: methanol at flow rate 0.3 mL/min, for UV-P, UV-326, and UV-329: 0–5.50 min 20% A, 9.00 min 1% A, 15.00 min 1%, A 15.50 min 20% A, and 18.00 min 70% A; for UV-327 and UV-328: 0–5.50 min 20% A, 9.00 min 1% A, 15.00 min 1% A 15.50 min 20% A, 18.00 min 70% A, 24.00 min 20% A, 29.00 min 1% A, 35.00 min 1% A 35.50 min 20% A, and 38.00 min 70% A.

Optimization of the benzotriazole oxidation process was made in Statistica 13.1 software (Tibco Software Inc., Palo Alto, CA, USA), using the Central Composite Design (CCD) technique from the design of experiments (DoE) method.

### 3.2. Procedure of Ultrasound-Assisted Emulsification Microextraction

The efficiency of BUVs’s extraction, using various organic solvents was tested. For this purpose, the ultrasound-assisted emulsification microextraction (USAEME) process was performed using chlorobenzene, chloroform, and toluene as extraction solvents. Additionally, USAEME, with solidification of the floating organic drop method (SFOD), using 1-undecanol and hexadecane, was used. Aliquots, of 5 mL of a water sample containing benzotriazoles, were placed in 10 mL glass centrifuge tubes. Then, 80 μL of extractant was added to the water sample and mixed. Immediately after, the tube was immersed in a Sonorex Digitec 102H ultrasonic water bath, Bandelin (Germany). Extractions were performed at 42 kHz of ultrasound frequency and 230 W of power, for 10 min at room temperature. The solvent volume of 80 µL and extraction time of 10 min were considered optimal, based on preliminary tests. Emulsions were disrupted by centrifugation, at 4000 rpm for 5 min, in an MPW-M UNIVERSAL Med. Instruments (Poland) laboratory centrifuge. Then, the organic phase settled at the bottom of the conical tube. After centrifugation, the organic layer was collected, using a 100 μL Agilent Technologies (USA) syringe and transferred into a 150 μL microvial with integrated insert. Then, extracts were subjected to GC-MS analysis. Chromatogram and mass spectra of tested benzotriazoles are shown in Appendix A. Results obtained for UV stabilizers indicate that the optimal extraction solvent is chlorobenzene (Appendix A), and this extractant was, then, used in all subsequent experiments.

### 3.3. Degradation Experiments

The preliminary degradation experiments of benzotriazoles were carried out in glass beakers, by mixing certain volumes of metal ions and benzotriazoles mixture, with an initial concentration of 500 µg/L. The pH of the mixture was adjusted, by adding a few microliters of NaOH (0.5 mol/L). Then, an appropriate volume of peracetic acid working solution was added to initiate the reaction. The oxidation reaction was carried out for 30 min, with continuous stirring of the solution at 700 rpm. Then, 800 µL 20% sodium thiosulfate was added, to stop the reaction. Quenching experiments were conducted by adding 0.5 mol/L TBA, 0.1 mol/L NaN_3_, or 0.01 mol/L 1,4-BQ into reaction solutions, before initiating a reaction. The experiment was carried out in dark conditions, to prevent the PAA from being activated by light.

## 4. Conclusions

For the first time, a procedure for advanced oxidation of benzotriazole UV filters, using peracetic acid activated with d-electron metal ions, was developed. A CCD-based chemometric approach was used to optimize of the oxidation process. At pH = 4.5, [PAA]_0_ = 25 mg/L and [Fe^2+^]_0_ = 6 × 10^−4^ mol/L as well as [PAA]_0_ = 40 mg/L and [Co^2+^]_0_ = 6 × 10^−4^ mol/L, the best oxidation efficiency of benzotriazoles was achieved. The effectiveness of the oxidizing system depends on the ratio of the concentration of peracetic acid and the activator, and, also, on the type of oxidized compound. Nevertheless, a slight advantage of the Fe^2+^/PAA system, over the system containing Co^2+^ ions as the PAA activator, is noticeable. Iron ions generate more reactive hydroxyl radicals and, therefore, increase the rate and efficiency of the oxidation reaction. Conducting experiments with the use of TBA, NaN_3_, and 1,4-BQ confirmed the earlier reports on the dominant role of ^•^OH radicals in the Fe^2+^/PAA process, and the influence of CH_3_C(O)OO^•^ and CH_3_C(O)O^•^ radicals on the oxidative activity of the Co^2+^/PAA system. This work extends the existing knowledge on the use of chemical oxidants to remove persistent pollutants from water matrices. The conducted research contributes significantly to the research on the oxidation processes in peracetic acid/d-electron metal ions systems, as the literature to date concerns only a few compounds exposed to these systems.

## Figures and Tables

**Figure 1 molecules-27-03349-f001:**
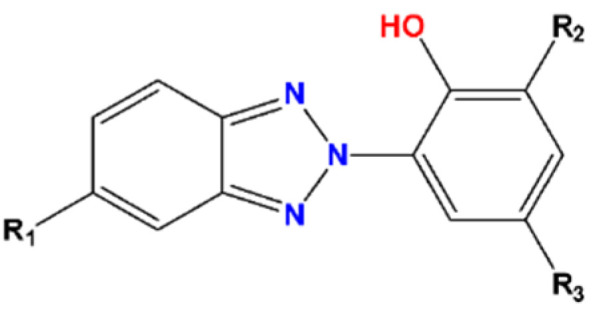
Benzotriazole UV stabilizer.

**Figure 2 molecules-27-03349-f002:**
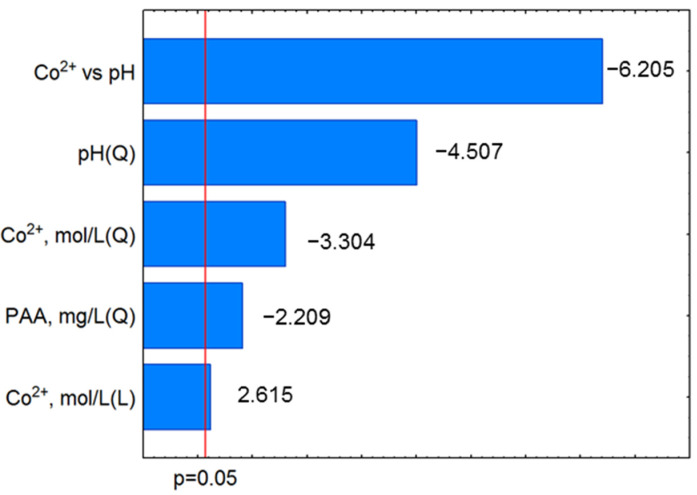
Absolute values of the standardized effects of UV-326 degradation in Co^2+^/PAA process.

**Figure 3 molecules-27-03349-f003:**
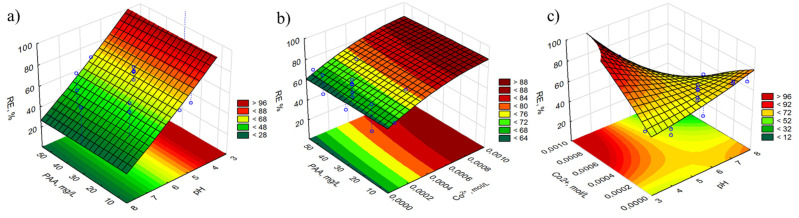
Response surface plots of RE%, as the function of two independent variables: (**a**) C_PAA_ and pH, (**b**) C_PAA_ and C_Co2+_, and (**c**) C_Co2+_ and pH. Conditions: [PAA]_0_ = 40 mg/L, [Co^2+^]_0_ = 8 × 10^−4^ mol/L, pH = 4.5.

**Figure 4 molecules-27-03349-f004:**
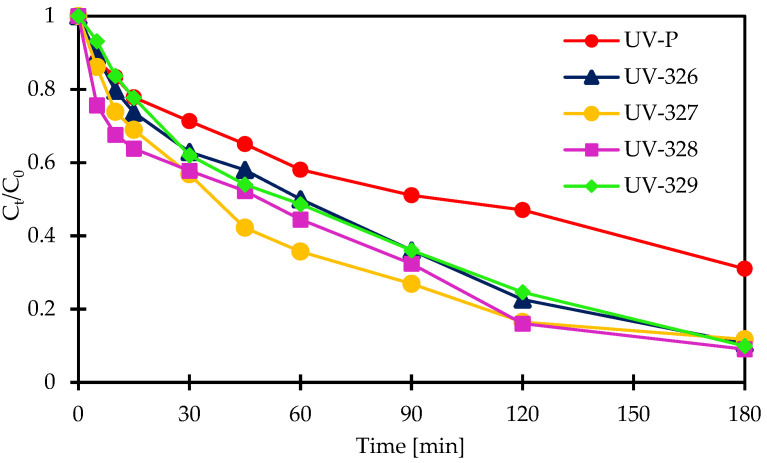
Kinetics of BUVs’s degradation in Fe^2+^/PAA process. Reaction conditions: [PAA]_0_ = 25 mg/L, [Fe^2+^]_0_ = 6 × 10^−4^ mol/L, initial pH 4.5.

**Figure 5 molecules-27-03349-f005:**
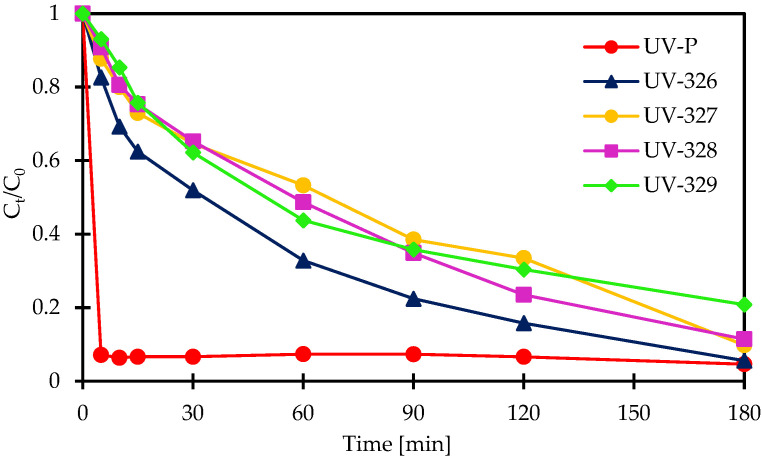
Kinetics of BUVs’s degradation in Co^2+^/PAA process. Reaction conditions: [PAA]_0_ = 40 mg/L, [Co^2+^]_0_ = 8 × 10^−4^ mol/L, initial pH 4.5.

**Figure 6 molecules-27-03349-f006:**
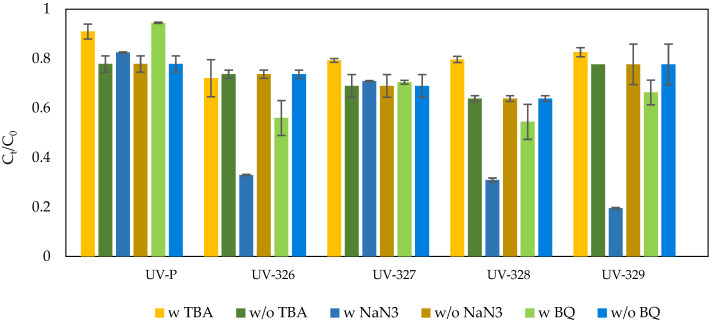
Effect of radical scavengers on UV stabilizers’ degradation in the Fe^2+^/PAA process (w”—reaction with radical quencher, w/o”—reaction without radical quencher).

**Figure 7 molecules-27-03349-f007:**
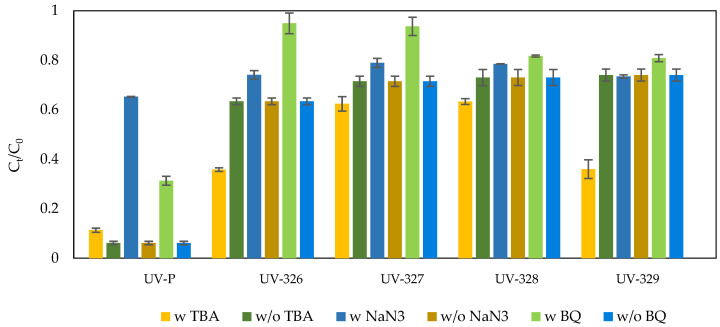
Effect of radical scavengers on UV stabilizers’ degradation in the Co^2+^/PAA process (w”—reaction with radical quencher, w/o”—reaction without radical quencher).

**Figure 8 molecules-27-03349-f008:**
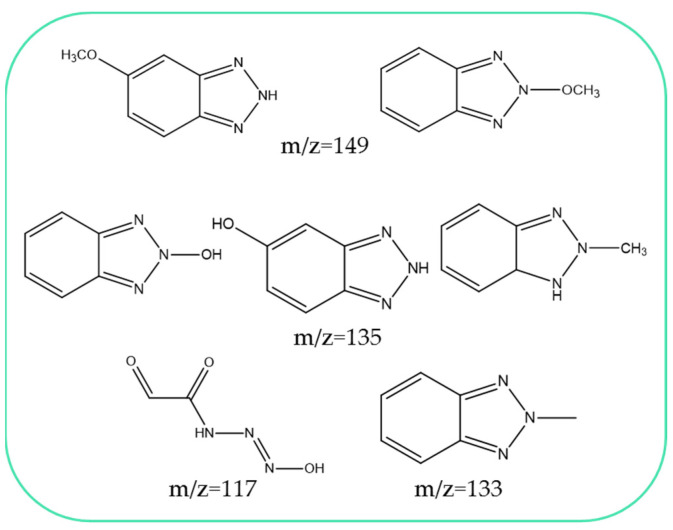
Proposed structures of benzotriazole UV stabilizers’ oxidation products.

**Table 1 molecules-27-03349-t001:** Concentration of target benzotriazoles in environmental and biological samples.

Sample	Location	UV–P	UV–326	UV–327	UV–328	UV–329	Determination Method	Ref.
Breast milk (ng/g lipid wt.)	South Korea	19.2	1.77	10	64.3	4.54	GC-MS	[32]
	Japan	21	0.08	n.d.	0.2	3.8	UHPLC-MS/MS	[33]
	Philippines	16/71 ^a^	34/64	n.d.	2.4/1.9	n.d.		
	Vietnam	91/3.9/32	0.53/n.d./2.1	n.d./n.d./1.6	0.9/0.48/0.47	9.6/2.6/6		
Mussels (ng/g)	Asia-Pacific coastal waters	–	150	68	130	–	GC-MS	[36]
WWTP (ng/L)	China	9.9–37.1 (7.2–15.9) ^b^	–	–	2.6–2.9 (0.60)	3.8	LC-MS/MS	[16]
Rivers in India (ng/L)	Water	0.2–2.3	1.5–3.7	3.3–4.3	0.5–3.4	8.1–13.7	GC-MS	[21]
	Sediment	0.1–0.3	0.2–0.5	0.6–2	0.2–0.9	0.9–1.41		
	Fish	2.2–6.9	0.6–1.6	1.0–3.2	0.2–1.6	3.0–7.4		
House dust (ng/L)	Philippines	–	53/6.2	28/10	50/18	–	UHPLC-ESI-MS/MS	[24]
Blood plasma of water animals (pg/g)	North America	–	–	–	240–776	<640	UPLC-MS/MS	[22]

^a^—the concentration after the slash applies to different locations; ^b^—concentration in influent (effluent).

**Table 2 molecules-27-03349-t002:** Acute toxicity of selected benzotriazoles on living organisms.

BUVs	Route	Living Organism	Acute Toxicity LD_50_/LC_50_	Ref.
UV-P	oral	Freshwater crustacean (*Daphnia pulex*)	>10 mg/L	[49]
	oral	mice	>5–>10 g/kg	[50]
	oral	rats	>15 g/kg	
	oral	rats	>5 g/kg	
	inhalation	rats	1420 mg/m^3^	
	dermal	rabbits	>2 g/kg	
	dermal	Guinea pigs	>3 g/kg	
UV-328	oral	Rat	7750 mg/kg	[51]
	inhalation	Rat	400 mg/m^3^	
	dermal	rabbit	1100 mg/kg	
	direct	Algae *Raphidocelis subcapitata*	EC_50_ > 0.016 mg/L	[52]

**Table 3 molecules-27-03349-t003:** The three-factor CCD matrix, with the experimental and predicted removal efficiency values for UV-326 degradation.

	[PAA]_0_ (mg/L)	[Me^2+^]_0_ (mol/L)	PAA/Fe^2+^ System			PAA/Co^2+^ System		
			pH	RE% (exp.)	RE% (pred.)	pH	RE% (exp.)	RE% (pred.)
1	45	3.45 × 10^−4^	4.6	91.01	100.00	7	57.35	63.37
2	45	3.45 × 10^−4^	3.4	94.13	97.24	4	76.25	77.84
3	45	1.45 × 10^−5^	4.6	36.99	55.25	7	73.69	72.96
4	45	1.45 × 10^−5^	3.4	33.35	46.27	4	55.32	60.78
5	15	3.45 × 10^−4^	4.6	97.44	100.00	7	66.91	64.61
6	15	3.45 × 10^−4^	3.4	92.99	97.24	4	75.48	79.08
7	15	1.45 × 10^−5^	4.6	20.10	55.25	7	75.22	74.20
8	15	1.45 × 10^−5^	3.4	23.84	46.27	4	49.54	62.02
9	55	7 × 10^−5^	4	78.68	66.76	5.5	67.55	66.21
10	5	7 × 10^−5^	4	93.51	66.76	5.5	65.49	68.28
11	30	1 × 10^−3^	4	90.15	87.04	5.5	65.51	64.61
12	30	5 × 10^−6^	4	41.04	53.29	5.5	55.44	66.56
13	30	7 × 10^−5^	5	93.50	61.48	8	64.78	76.62
14	30	7 × 10^−5^	3	57.71	46.51	3	58.76	63.78
15	30	7 × 10^−5^	4	75.96	66.76	5.5	69.46	67.68
16	30	7 × 10^−5^	4	76.79	66.76	5.5	77.01	67.68
17	30	7 × 10^−5^	4	71.59	66.76	5.5	77.37	67.68
18	30	7 × 10^−5^	4	72.02	66.76	5.5	77.74	67.68
19	30	7 × 10^−5^	4	72.57	66.76	5.5	81.38	67.68
20	30	7 × 10^−5^	4	78.93	66.76	5.5	76.76	67.68

exp.—experimental; pred.—predicted.

**Table 4 molecules-27-03349-t004:** ANOVA results for UV-326 removal efficiency from CCD.

Source of Variation	Sum of Squares	DF	Mean Square	*F*-Value	*p*-Value
PAA concentration (square)	128.572	1	128.5723	8.46121	0.033446
Co^2+^ concentration (linear)	103.937	1	103.9369	6.83998	0.047364
Co^2+^ concentration (square)	165.885	1	165.8846	10.91670	0.021380
pH (square)	308.624	1	308.6242	20.31025	0.006360
Co^2+^ concentration-pH interactions	585.017	1	585.0169	38.49937	0.001588
Lack of fit	507.354	9	56.3727		
Pure error	75.977	5	15.1955		
Total	1610.347	19			
R^2^ = 0.63776	R^2^ (adjusted) = 0.50839		*p* < 0.05 is considered as significant		

**Table 5 molecules-27-03349-t005:** Determination coefficients (R^2^), first-order constant (k), and half-life time (t_1/2_), of BUVs’s removal by Fe^2+^, for Co^2+^/PAA-based oxidation.

Compound	Fe^2+^/PAA Process			Co^2+^/PAA Process		
	R^2^	k (min^−1^)	t_1/2_ (min)	R^2^	k (min^−1^)	t_1/2_ (min)
UV-P	0.972	0.0059	117.48	–	–	–
UV-326	0.992	0.0118	58.74	0.992	0.0150	46.21
UV-327	0.967	0.0166	41.76	0.953	0.0107	64.78
UV-328	0.976	0.0125	55.45	0.997	0.0116	59.75
UV-329	0.991	0.0121	57.28	0.960	0.0088	78.77

## Data Availability

Not applicable.

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
