# Peer review of "Degradation of Benzotriazole UV Stabilizers in PAA/d-Electron Metal Ions Systems—Removal Kinetics, Products and Mechanism Evaluation"

_molecules, 2022, doi:10.3390/molecules27103349_

Round 1

Reviewer 1 Report

In this manuscript, Kiejza et al. reported degradation of benzotriazole UV stabilizers in PAA/d-electron metal ions systems–removal kinetics, products and mechanism. There are some interesting results related to the manuscript. However, it is recommended that the article not be accepted in its current form. Minor revision is needed before publication.

  1. In this manuscript, authors presents the use of peracetic acid in combination with d-electron metal ions (Fe2+, Co2+) for the chemical oxidation of five UV filters from the benzotriazole group: 2-(2-hydroxy-5-methylphenyl)benzotriazole (UV-P), 2-tert-butyl-6-(5-chloro-2H-benzotriazol-2-yl)-4-methylphenol (UV-326), 2,4-di-tert-butyl-6-(5-chloro-2H-benzotriazol-2-yl)phenol (UV-327), 2-(2H-benzotriazol-2-yl)-4,6-di-tert-pentylphenol (UV-328) and 2-(2H-benzotriazol-2-yl)-4-(1,1,3,3-tetramethylbutyl)phenol (UV-329). The oxidation procedure has been optimized based on the Design of Experiment (DoE) methodology. The oxidation of benzotriazoles follows first order kinetics. The author needs to explain the advantages and innovations of this work more clearly.
  2. The tables are suggested to re-organized to make sure a table appears on a page.
  3. At least three times experiments should be conducted, error bars are suggested to added in some of the figures, for example, Figure 6, Figure 7.
  4. The recent related reference are suggested to read, such as ACS Appl. Mater. Interfaces 2020, 12, 31, 34999–35010, 10.1016/j.cej.2020.124780. Journal of Colloid and Interface Science, 2022, 608: 164-174.

Author Response

I am grateful for the careful evaluation of the manuscript and comments improving the quality of the work presented. As authors, we have carefully analyzed all the tips. We took all of the comments very seriously and made changes to our work on their basis. The changes made are marked and their location is indicated.

#Reviewer 1

In this manuscript, Kiejza et al. reported degradation of benzotriazole UV stabilizers in PAA/d-electron metal ions systems–removal kinetics, products and mechanism. There are some interesting results related to the manuscript. However, it is recommended that the article not be accepted in its current form. Minor revision is needed before publication.

  1. In this manuscript, authors presents the use of peracetic acid in combination with d-electron metal ions (Fe2+, Co2+) for the chemical oxidation of five UV filters from the benzotriazole group: 2-(2-hydroxy-5-methylphenyl)benzotriazole (UV-P), 2-tert-butyl-6-(5-chloro-2H-benzotriazol-2-yl)-4-methylphenol (UV-326), 2,4-di-tert-butyl-6-(5-chloro-2H-benzotriazol-2-yl)phenol (UV-327), 2-(2H-benzotriazol-2-yl)-4,6-di-tert-pentylphenol (UV-328) and 2-(2H-benzotriazol-2-yl)-4-(1,1,3,3-tetramethylbutyl)phenol (UV-329). The oxidation procedure has been optimized based on the Design of Experiment (DoE) methodology. The oxidation of benzotriazoles follows first order kinetics. The author needs to explain the advantages and innovations of this work more clearly.

Relevant passages have been added to the end of Intoduction (lines 104-107) and to the end of Conclusions (lines 382-386):

“The aim of the presented work was to develop a new approach to the removal of BUVs as persistent environmental pollutants from aqueous solutions, based on the use of advanced oxidation. The literature review shows that, until now, no attempt has been made to use chemical oxidants to degrade and dispose of BUVs.”

“This work extends the existing knowledge on the use of chemical oxidants to remove persistent pollutants from water matrices. The conducted research contributes signifi-cantly to the research on the oxidation processes in peracetic acid/d-electron metal ions systems, as the literature to date concern only a few compounds exposed to these systems.”

  1. The tables are suggested to re-organized to make sure a table appears on a page.

Tables have been reorganized to appear on one page. In subsection 2.4. “Benzotriazole UV stabilizers degradation products” Figure 8 was inserted instead of Table 6. This allowed for a clearer presentation of the results of the experiment. The other oxidation products mentioned in this subsection were transferred to Supplementary Materials and designated as Table S11. Therefore, in the subsection Supplementary Materials in the manuscript, the following sentence was added: "Table S11: Proposed structures of benzotriazole UV stabilzers oxidation products".

  1. At least three times experiments should be conducted, error bars are suggested to added in some of the figures, for example, Figure 6, Figure 7.

All experiments were performed at least three times. In figures 6 and 7 error bars have been added.

  1. The recent related reference are suggested to read, such as ACS Appl. Mater. Interfaces 2020, 12, 31, 34999–35010, 10.1016/j.cej.2020.124780. Journal of Colloid and Interface Science, 2022, 608: 164-174.

The authors carefully looked at the proposed works. Undoubtedly, they broadened our knowledge of modern water and wastewater treatment technologies. Due to the large discrepancy with the topics we discussed, we decided not to include them in the publication.

Reviewer 2 Report

The manuscript entitled “Degradation of benzotriazole UV stabilizers in PAA/d-electron metal ions systems–removal kinetics, products and mechanism evaluation, describes the use of peracetic acid in combination with d-electron metal ions (Fe2+, Co2+) for the chemical oxidation of five UV filters from the benzotriazole group which eliminates adverse effects on living organisms.

Please include the reference Journal of Photochemistry & Photobiology, B: Biology 230 (2022) 112444

Author Response

I am grateful for the careful evaluation of the manuscript and comments improving the quality of the work presented. As authors, we have carefully analyzed all the tips. We took all of the comments very seriously and made changes to our work on their basis. The changes made are marked and their location is indicated.

#Reviewer 2

The manuscript entitled “Degradation of benzotriazole UV stabilizers in PAA/d-electron metal ions systems–removal kinetics, products and mechanism evaluation, describes the use of peracetic acid in combination with d-electron metal ions (Fe2+, Co2+) for the chemical oxidation of five UV filters from the benzotriazole group which eliminates adverse effects on living organisms.

Please include the reference Journal of Photochemistry & Photobiology, B: Biology 230 (2022) 112444.

Cited reference was placed at the end of section 1. "Introduction" as a reference number 74. Accordingly, there was a shift in the numbering of the references: reference 74 moved to position 75, 75 to 76 and so did the following until the end of the manuscript.
